# CogVLM: Visual Expert for Pretrained Language Models

**Weihan Wang**[*1,2]**, Qingsong Lv**[*1]**, Wenmeng Yu**[1]**, Wenyi Hong**[1,2]**, Ji Qi**[1,2]**, Yan Wang**[1]**,
Junhui Ji**[1]**, Zhuoyi Yang**[1,2]**, Lei Zhao**[1]**, Xixuan Song**[1,2]**, Jiazheng Xu**[1,2]**, Keqin Chen**[1]**,
Bin Xu**[2]**, Juanzi Li**[2]**, Yuxiao Dong**[†2]**, Ming Ding**[†1]**, Jie Tang**[†2]

[1]Zhipu AI    [2]Tsinghua University
ming.ding@zhipuai.cn    {yuxiaod, jietang}@tsinghua.edu.cn

## Abstract

We introduce CogVLM, a powerful open-source visual language foundation model. Different from the popular *shallow alignment* method which maps image features into the input space of language model, CogVLM bridges the gap between the frozen pretrained language model and image encoder by a trainable visual expert module in the attention and FFN layers. As a result, CogVLM enables a deep fusion of vision language features without sacrificing any performance on NLP tasks. CogVLM-17B achieves state-of-the-art performance on 15 classic cross-modal benchmarks, including 1) image captioning datasets: NoCaps, Flicker30k, 2) VQA datasets: OKVQA, ScienceQA, 3) LVLM benchmarks: MM-Vet, MMBench, SEED-Bench, LLaVABench, POPE, MMMU, MathVista, 4) visual grounding datasets: RefCOCO, RefCOCO+, RefCOCOg, Visual7W. Codes and checkpoints are available at Github.

## 1   Introduction

Vision language models are versatile and powerful. Many vision and cross-modality tasks can be formulated as next token prediction, e.g., image captioning [Agrawal et al., 2019], visual question answering [Antol et al., 2015], visual grounding [Yu et al., 2016] and even segmentation [Chen et al., 2022a]. Useful abilities like in-context learning [Tsimpoukelli et al., 2021, Sun et al., 2023b, Alayrac et al., 2022] also emerge along with the improvement of downstream tasks when scaling up VLMs. However, to train a large language model is already non-trivial, and it is more challenging to train a VLM from scratch with the same NLP performance as well-trained pure language models like LLaMA2 [Touvron et al., 2023]. Therefore, it is natural to investigate how to train a VLM from an off-the-shelf pretrained language model.

The popular *shallow alignment* methods represented by InstructBLIP [Li et al., 2023b] and MiniGPT-4 [Zhu et al., 2023] connect a frozen pretrained vision encoder and language model via a trainable Q-Former or a linear layer, mapping the image features into the input embedding space of the language model. This method converges rapidly, but its performance is noticeably inferior to that of LLaVA-1.5 with trainable language parameters, despite their model sizes and training datasets being almost identical.

The primary challenge in the performance of shallow alignment methods within VLMs can be attributed to the lack of deep fusion between visual and linguistic data. Shallow alignment methods struggle because they rely on 'frozen' language model weights, which are intrinsically trained to

---

[*]WW & QL contributed equally.

[†]Corresponding authors

[‡]Work done when WW, WH, JQ, ZY, XS, JX and KC interned at Zhipu AI.

38th Conference on Neural Information Processing Systems (NeurIPS 2024).

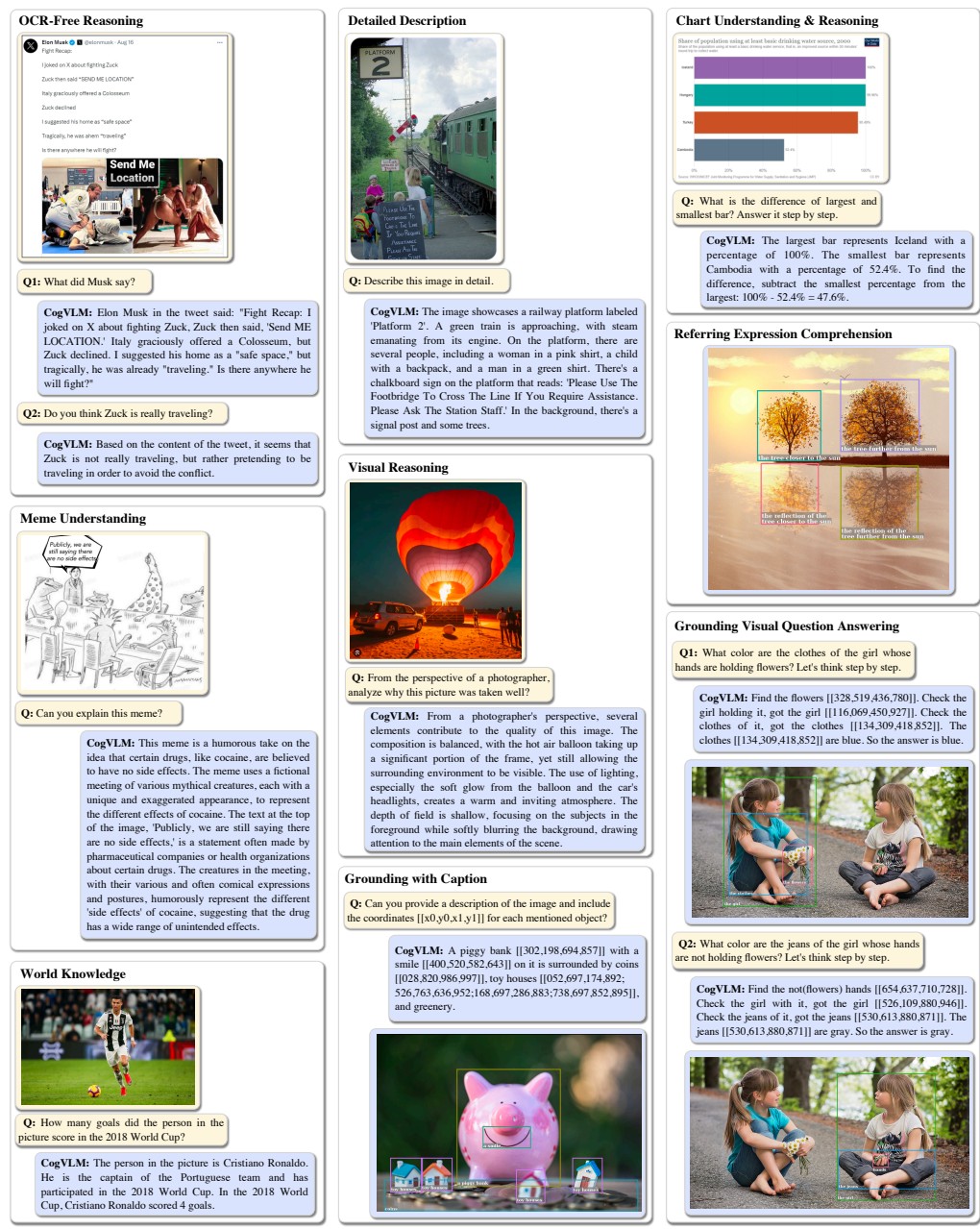

Figure 1: Samples generated by CogVLM.

process text tokens. This presents a significant mismatch issue, as visual features lack a direct equivalent in the textual input space. Consequently, when these visual features undergo multi-layer transformations, they tend to deviate from the expected input distribution of the deeper language model layers. This misalignment is particularly evident in tasks like image captioning, where the specificity of a task – such as writing style and caption length – can only be superficially encoded into visual features through shallow methods.

A common strategy, as seen in PaLI [Chen et al., 2022b] and Qwen-VL [Bai et al., 2023], involves direct training of LLM during the pre-training or supervised fine-tuning (SFT) phase. However, this approach can compromise the models' generalizability, particularly for tasks focused on textual outputs. Conventionally, LLMs are pretrained on extensive text-only datasets [Raffel et al., 2020], leading to a significant divergence in data distribution when compared to image-text pair datasets like LAION [Schuhmann et al., 2022] and COYO [Byeon et al., 2022]. This shift often results in catastrophic forgetting, a phenomenon where the model's proficiency in its original domain deteriorates. This issue is evident in Figure 2, which shows a marked decline in MMLU [Hendrycks et al., 2020] score as the model becomes more attuned to the LAION dataset, thus validating our

hypothesis. This trend is not isolated; similar effects have been observed in models like PaLM-E [Driess et al., 2023] and Flamingo [Alayrac et al., 2022]. For instance, adapting an 8B parameter language model for VLM pretraining can lead to an 87.3% reduction in natural language generation (NLG) performance [Driess et al., 2023].

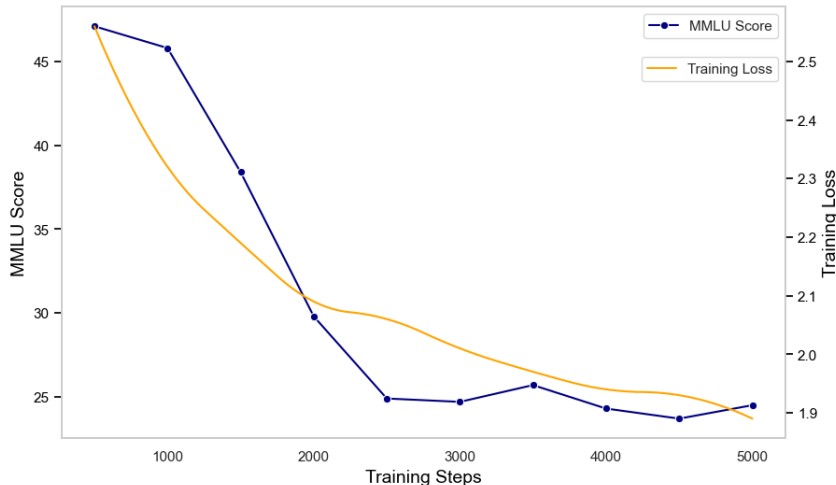

Figure 2: **MMLU score and training loss over multimodal pretraining phase.** When directly training the language part of the VLM using the LAION dataset, the model's score on the pure text dataset MMLU rapidly decreases, dropping to 24.9 at 2500 steps.

The discussion above raises an important question: is it possible to retain the NLP capabilities of the large language model while adding top-notch visual understanding abilities to it?

CogVLM gives a "*yes*" answer. CogVLM instead adds a trainable *visual expert* to the language model. In each layer, the image features in the sequence use a new QKV matrix and MLP layer with the text features. Visual expert doubles the number of parameters while keeping the FLOPs the same. Since all the parameters in the original language model are fixed, the behaviors are the same as in the original language model if the input sequence contains no image. This inspiration arises from the comparison between P-Tuning [Liu et al., 2023f] and LoRA [Hu et al., 2021] in efficient finetuning, where p-tuning learns a task prefix embedding in the input while LoRA adapts the model weights in each layer via a low-rank matrix. As a result, LoRA performs better and more stable. A similar phenomenon might also exist in VLM, because in the shallow alignment methods, the image features act like the prefix embedding in P-Tuning.

Our contributions in this work are as follows:

- We introduce the CogVLM model, which deeply integrates visual and linguistic features while retaining the full capabilities of a pretrained large language model. CogVLM-17B, trained from Vicuna-7B, achieves state-of-the-art across 17 classic cross-modal benchmarks.

- Through extensive ablation studies, we validated the effectiveness of our proposed visual expert module and the importance of deep fusion. We further delved into multiple critical factors in multimodal pertaining, including the scale of visual encoder, variants of attention mask, the most impactful parameters in VLMs, and the necessity of incorporating self-supervised image loss, etc.

- We have made the weights of CogVLM and the dataset used in the SFT phase available to the public. We anticipate that the open sourcing of CogVLM will significantly contribute to the research and industrial application of visual understanding.

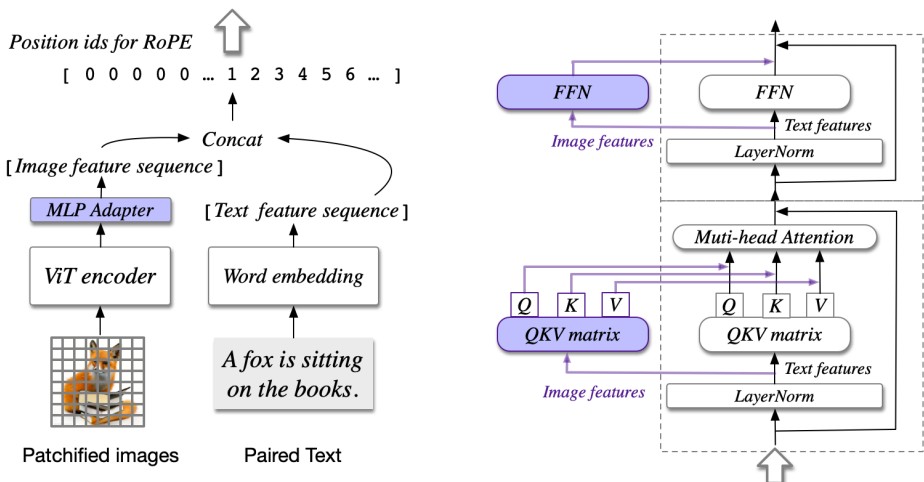

(a) The input of visual language model        (b) The visual expert built on the language model

Figure 3: **The architecture of CogVLM.** (a) The illustration about the input, where an image is processed by a pretrained ViT and mapped into the same space as the text features. (b) The Transformer block in the language model. The image features have a different QKV matrix and FFN. Only the purple parts are trainable.

## 2 Method

### 2.1 Architecture

CogVLM model comprises four fundamental components: a vision transformer (ViT) encoder, an MLP adapter, a pretrained large language model (GPT), and a visual expert module. Figure 3 shows an overview of the CogVLM architecture. The components' design and implementation details are provided below:

**ViT encoder**. We utilize pretrained EVA2-CLIP-E [Sun et al., 2023a] in CogVLM-17B. Note that the final layer of ViT encoder is removed because it specializes in aggregating the [CLS] features for contrastive learning.

**MLP adapter**. To map the output of ViT into the same space as the text features from word embedding, we use an MLP adapter, a two-layer MLP (SwiGLU [Shazeer, 2020]). For implementation convenience, all image features share the same position id in the language model.

**Pretrained large language model**. CogVLM's model design is compatible with any off-the-shelf GPT-style pretrained large language model. Specifically, CogVLM-17B adopts Vicuna1.5-7B [Chiang et al., 2023] for further training. A causal mask is applied to all the attention operations, including the attention between image features.

**Visual expert module**. We add a visual expert module to each layer to enable deep visual-language feature alignment. Specifically, the visual expert module in each layer consists of a QKV matrix and an MLP in each layer. The shapes of the QKV matrix and MLP are identical to those in the pretrained language model and initialized from them. The motivation is that each attention head in the language model captures a certain aspect of semantic information, while a *trainable* visual expert can transform the image features to align with the different heads, therefore enabling deep fusion.

Formally, suppose that the input hidden states of an attention layer are $X \in \mathbb{R}^{B \times H \times (L_I + L_T) \times D}$, where $B$ is the batch size, $L_I$ and $L_T$ are the lengths of image and text sequences, $H$ is the number of attention heads, and $D$ is the hidden size. In the attention with visual expert, $X$ is first split as image hidden states $X_I$ and text hidden states $X_T$, and the attention is computed as:

$$\text{Attention}(X, W_I, W_T) = \text{softmax}(\frac{\text{Tril}(QK^T)}{\sqrt{D}})V, \tag{1}$$

$$Q = \text{concat}(X_I W_I^Q, X_T W_T^Q), \tag{2}$$

$$K = \text{concat}(X_I W_I^K, X_T W_T^K), \tag{3}$$

$$V = \text{concat}(X_I W_I^V, X_T W_T^V), \tag{4}$$

where $W_I$, $W_T$ are the QKV matrices of the visual expert and original language model, and $\text{Tril}(\cdot)$ means lower-triangular mask. The visual expert in FFN layers performs similarly,

$$\text{FFN}(X) = \text{concat}(\text{FFN}_I(X_I), \text{FFN}_T(X_T)), \tag{5}$$

where $\text{FFN}_I$ and $\text{FFN}_T$ are the FFN of the visual expert and original language model.

**Position embedding.** In the RoPE within LLM, we allow all visual tokens to share a single position id, as they already encapsulate positional information when inputted into the ViT. This approach mitigates the impact of remote attenuation between tokens in the LLM. Given that an image can occupy hundreds to thousands of tokens, and a typical input sequence is structured as *'<image embed> query'*, using conventional positional encoding would result in excessively lengthy encoding sequences. Moreover, it would lead the query to focus more on the image sequences closer to it, namely the lower part of an image.

## 2.2 Pretraining

**Data.** The image-text pairs for pretraining are all publicly available, including LAION-2B and COYO-700M. After removing the broken URLs, NSFW images, images with noisy captions, images with political bias and images with an aspect ratio $> 6$ or $< 1/6$, about 1.5B images are left for pretraining.

We also crafted a visual grounding dataset of 40M images. Each noun in the image caption is associated with bounding boxes to indicate the positions in the image. The construction process basically follows [Peng et al.], which extracts nouns via spaCy [Honnibal and Johnson, 2015] and predicts the bounding boxes using GLIPv2 [Zhang et al., 2022]. The image-text pairs are sampled from LAION-115M, a subset of LAION-400M filtered by [Li et al., 2023b]. We filter and retain a subset of 40 million images to ensure that over 75% of images contain at least two bounding boxes.

**Training.** The first stage of pretraining is for *image captioning loss*, i.e. next token prediction in the text part. We train the CogVLM-17B model on the 1.5B image-text pairs introduced above for 120,000 iterations with a batch size of 8,192. The second stage of pretraining is a mixture of image captioning and Referring Expression Comprehension (REC). REC is a task to predict the bounding box in the image given the text description of an object, which is trained in the form of VQA, i.e., *Question: Where is the* object*?* and *Answer:* $[[x_0, y_0, x_1, y_1]]$. Both $x$ and $y$ coordinates range from 000 to 999, meaning the normalized position in the image. We only consider the loss of the next token prediction in the "Answer" part. We pretrain the second stage for 60,000 iterations with a batch size of 1,024 on the text-image pairs and visual grounding datasets introduced above. During the final 30,000 iterations, we change the input resolution from $224 \times 224$ to $490 \times 490$. The total number of trainable parameters is 6.5B.

## 2.3 Alignment

In the instruction alignment phase, we trained two generalist models: CogVLM-Chat and CogVLM-Grounding. CogVLM-Chat accepts natural language inputs and outputs, while CogVLM-Grounding accepts inputs and outputs with bounding boxes.

**CogVLM-Chat.** In our study, we integrated data from a variety of open-source visual question-answering datasets, including VQAv2 [Antol et al., 2015], OKVQA [Marino et al., 2019], TextVQA [Singh et al., 2019], OCRVQA [Mishra et al., 2019], ScienceQA [Lu et al., 2022], as well as datasets formatted as multi-turn dialogues such as LLaVA-Instruct [Liu et al., 2023c], LRV-Instruction [Liu et al., 2023a], LLaVAR [Zhang et al., 2023b]. We then conducted unified instruction-supervised fine-tuning (SFT) across these diverse datasets. The integrity and quality of SFT data are crucial; notably, the LLaVA-Instruct dataset, initially generated through a language-only GPT-4 pipeline, contained certain inaccuracies. We meticulously corrected these errors through manual inspection and annotation to ensure data quality.

VQA datasets typically feature concise, often one-word answers, contrasting with the dialogue datasets that provide detailed responses with extensive reasoning. To accommodate this variability,

Table 1: **Performance on Image Captioning benchmarks.** All tasks use CIDEr as the evaluation metric. OOD refers to out-of-domain test set. Karp. refers to the Karpathy test split.

| Method | Train Data | NoCaps val | | NoCaps test | | Flickr | COCO | TextCaps |
|---|---|---|---|---|---|---|---|---|
| | | OOD | overall | OOD | overall | Karp. | Karp. | test |
| Human | - | 95.7 | 87.1 | 91.6 | 85.3 | - | - | 125.1 |
| VinVL [Zhang et al., 2021] | 8.9M | 83.8 | 94.3 | 78.0 | 92.5 | - | 130.8 | - |
| SimVLM [Wang et al., 2021] | 1.8B | 115.2 | 112.2 | 109.5 | 110.3 | - | 143.3 | - |
| CoCa [Yu et al., 2022] | 4.8B | - | 122.4 | - | 120.6 | - | 143.6 | - |
| LEMON [Hu et al., 2022] | 2B | 120.2 | 117.3 | 110.1 | 114.3 | - | 139.1 | - |
| Flamingo [Alayrac et al., 2022] | 2.3B | - | - | - | - | 67.2 | 138.1 | - |
| Prismer [Liu et al., 2023d] | 12.7M | 113.5 | 112.9 | - | 110.8 | - | 136.5 | - |
| BLIP-2 [Li et al., 2023b] | 129M | 124.8 | 121.6 | - | - | - | 144.5 | - |
| InstructBLIP [Dai et al., 2023] | 129M | - | 123.1 | - | - | 82.4 | - | - |
| UniversalCap [Cornia et al., 2021] | 35M | 123.4 | 122.1 | 114.3 | 119.3 | - | 143.4 | - |
| GIT [Wang et al., 2022a] | 0.8B | 127.1 | 125.5 | 122.0 | 123.4 | 49.6 | 144.8 | 138.2 |
| GIT2 [Wang et al., 2022a] | 12.9B | 130.6 | 126.9 | 122.3 | 124.8 | 50.7 | 145.0 | 145.0 |
| Qwen-VL [Bai et al., 2023] | 1.4B | - | 121.4 | - | - | 85.8 | - | - |
| PaLI-17B [Chen et al., 2022b] | 1.6B | - | 127.0 | - | 124.4 | - | 149.1 | 135.4 |
| PaLI-X-55B [Chen et al., 2023b] | - | - | 126.3 | - | 124.3 | - | **149.2** | **147.0** |
| CogVLM (ours) | 1.5B | **132.6** | **128.3** | **128.0** | **126.4** | **94.9** | 148.7 | 144.9 |

we employed prompts formatted as *Question: Short answer:* for concise responses and *Question: Answer:* for extended discourse in the SFT phase.

During training, the model underwent 6000 iterations with a learning rate of 1e-5 and a batch size of 1024. To enhance and ensure the stability of the training, we activated the visual encoder's parameters and adjusted its learning rate to be one-tenth of that used for the remaining training parameters.

**CogVLM-Grounding.** In order to endow our model with consistent, interactive visual grounding capabilities, we collect a high-quality dataset covering 4 types of grounding data: (1) **Grounded Captioning (GC)** - image captioning datasets where each noun phrase within the caption is followed by the corresponding referential bounding boxes; (2) **Referring Expression Generation (REG)** - image-oriented datasets that each bounding box in the image is annotated with a descriptive textual expression that accurately characterizes and refers to the content within the specific region; (3) **Referring Expression Comprehension (REC)** - text-oriented datasets that each textual description is annotated with multiple referential links associating the phrases with corresponding boxes; (4) **Grounded Visual Question Answering (GroundedVQA)** - VQA-style datasets where the questions may contain region references in a given image. The sources of grounding data are all publicly available, including Flickr30K Entities [Plummer et al., 2015], RefCOCO [Kazemzadeh et al., 2014, Mao et al., 2016, Yu et al., 2016], Visual7W [Zhu et al., 2016], VisualGenome [Krishna et al., 2017] and Grounded CoT-VQA [Chen et al., 2023a]. $[box]$ in this section is in the format of $[[x_0, y_0, x_1, y_1]]$.

It is noteworthy that the curated datasets exhibit a versatility of visual grounding capabilities, and many datasets can be adapted and repurposed across different tasks. For instance, grounded captioning datasets can be reformulated to suit REG and REC tasks. Taking the example of *"A man $[box_1]$ and a woman $[box_2]$ are walking together.",* this can be reframed into question answering pairs like *("Describe this region $[box_2]$.", "A woman.")* and *("Where is the man?", "$[box_1]$").* Similarly, REC datasets can be translated into REG tasks by switching the input and output, and vice versa. However, certain conversions might lead to ambiguities. For example, when presented with the isolated query "Where is another man?" from the caption "A man $[box_1]$ is running, while another man $[box_2]$ is looking.", the distinction between $[box_1]$ and $[box_2]$ becomes unclear, potentially leading to errors.

## 3 Experiments

To rigorously validate the superior performance and robust generalization of our base model, we conduct quantitative evaluations on an array of multi-modal benchmarks. These benchmarks can be categorized into three broad areas covering a comprehensive range of measurement[1]:

- **Image Captioning**. The main purpose of these tasks is to generate textual captions summarizing the major content of a given image. We utilize prominent datasets including

---

[1]Detailed summary of all benchmarks and corresponding metrics are available at Appendix A.2.

Table 2: **Generalist performance on VQA and LVLM benchmarks.** * donates the dataset has been trained during SFT stage. We compared with the latest state-of-the-art generalist models, including MiniGPT-4 [Zhu et al., 2023], IDEFICS-Instruct [Laurençon et al., 2023], OpenFlamingo [Awadalla et al., 2023], DreamLLM [Dong et al., 2023], InstructBLIP [Dai et al., 2023], Fuyu [Bavishi et al., 2023], Qwen-VL [Bai et al., 2023], LLaVA-1.5 [Liu et al., 2023b], InternLM-XComposer [Zhang et al., 2023a]mPLUG-Owl2 [Ye et al., 2023], SPHINX [Lin et al., 2023b], Emu2 [Sun et al., 2023b].

| Method | LLM | VQA | | | LVLM-Benchmark | | | | | | |
|---|---|---|---|---|---|---|---|---|---|---|---|
| | | VQAv2 | OKVQA | ScienceQA | MM-Vet | SEED | MMBench | LLaVA | POPE | MMMU | MathVista |
| MiniGPT-4 | Vicuna-7B | - | - | 39.6 | 22.1 | 47.4 | 23.0 | 45.1 | - | - | 23.1 |
| IDEFICS-Instruct | LLaMA-65B | 37.4 | 36.9 | 61.8 | 39.7 | 53.2 | 54.5 | 56.9 | - | - | 26.2 |
| OpenFlamingo | MPT-7B | 53.0 | 38.3 | 44.8 | 24.8 | 42.7 | 5.7 | 34.2 | - | 26.3 | 18.6 |
| DreamLLM | Vicuna-7B | 56.6 | 44.3 | - | 35.9 | - | 49.9 | - | - | - | - |
| InstructBLIP | Vicuna-7B | - | - | 60.5 | 26.2 | 58.8 | 33.9 | 59.8 | 53.8 | - | 25.3 |
| Fuyu | Fuyu-8B | 74.2* | 60.6* | - | - | - | - | - | - | 27.4 | - |
| Qwen-VL-Chat | Qwen-7B | 78.2* | 56.6* | 68.8 | - | 65.4 | 61.8 | 67.7 | - | 32.9 | 33.8 |
| LLaVA-1.5 | Vicuna-7B | 78.5* | - | 66.8 | 30.5 | 58.6 | 64.3 | 60.7 | 85.9 | - | 23.6 |
| InternLM-XComposer | InternLM-7B | - | - | - | 35.2 | 66.9 | 74.4 | - | - | - | 29.8 |
| mPLUG-Owl2 | LLaMA2-7B | 79.4* | 57.7* | 68.7 | 36.2 | 64.1 | 64.5 | 25.0 | 86.2 | 32.1 | 25.3 |
| Unified-IO2 | UIO-2XXL | 79.4* | 55.5* | 86.2* | - | 65.6 | 71.5 | - | 87.7 | - | - |
| LLaVA-1.5 | Vicuna-13B | 80.0* | - | 71.6 | 35.4 | 61.6 | 67.7 | 64.6 | 85.9 | 33.6 | 26.1 |
| SPHINX-2k | LLaMA2 13B | 80.7* | 62.6* | 70.6 | 40.2 | 71.6 | 65.9 | - | 87.2 | 32.9 | 27.8 |
| Emu2-Chat | LLaMA-33B | **84.9*** | 64.8* | - | 48.5 | 62.8 | 63.6 | 56.4 | - | 34.1 | - |
| CogVLM-Chat | Vicuna-7B | 82.3* | **64.8*** | 91.2* | 51.1 | 72.5 | 77.6 | 77.8 | 87.9 | 41.1 | 34.5 |
| CogVLM-Chat | LLaMA3-8B | 83.4* | 64.1* | **92.5*** | **60.4** | **75.9** | **80.5** | **86.4** | **88.2** | **44.3** | **38.1** |

NoCaps [Agrawal et al., 2019], COCO [Lin et al., 2014], Flickr30K [Plummer et al., 2015], and TextCaps [Sidorov et al., 2020] for evaluation.

- **Visual Question Answering**. The VQA tasks require models to answer questions that may focus on distinct visual contents based on the given image. Our assessment covers diverse datasets, including VQAv2 [Antol et al., 2015], OKVQA [Marino et al., 2019] and ScienceQA [Lu et al., 2022].

- **LVLM Benchmarks**. LVLM benchmarks are primarily employed to assess the advanced capabilities of large multimodal models, such as object recognition and localization, OCR, visual description, and visual knowledge reasoning. We conduct multidimensional evaluations of the models on datasets including MM-Vet [Yu et al., 2023], MMBench [Liu et al., 2023g], SEED-Bench [Li et al., 2023a], LLaVA-Bench [Liu et al., 2023c], POPE [Li et al., 2023c], MMMU [Yue et al., 2023] and MathVista [Lu et al., 2023].

- **Visual Grounding**. Visual grounding involves a set of tasks that establish referential links between textual mentions in a sentence and specific regions in an image. We evaluate our model on the typical datasets, including Visual7w [Zhu et al., 2016], RefCOCO [Liu et al., 2017], RefCOCO+, and RefCOCOg to ensure completeness.

## 3.1 Image Captioning

We evaluate the image captioning capability of our pretrained base model on the aforementioned four benchmarks. In a zero-shot evaluation on the Nocaps and Flickr datasets, we assess the precision of our model in describing long-tail visual concepts. Additionally, we present results from finetuning on the COCO and TextCaps datasets.

The detailed performance is shown in Table 1. Overall, our model achieves the SOTA or compatible performance across the board. Specifically, on the NoCaps benchmark, our base model outperforms the previous best method, GIT2, across four splits with a maximum of 5.7 points in the out-domain set while only consuming 10% of the pretraining data (1.5B vs 12.9B). On the Flickr benchmark, our model achieves a SOTA score of 94.9 surpassing the concurrently released Qwen-VL model by 9.1 points. These results demonstrate the remarkable capability and robustness of our pretrained model on the image captioning task. We also evaluate our model on the COCO [Lin et al., 2014] and TextCaps, where the latter is specifically designed to integrate the textual information of the given image into captions. Though training without the dedicated OCR data, encouragingly, our base model

reveals a significant text-reading ability and obtains a competitive performance with PaLI-X-55B, and outperforms the previous best model of the same scale, PaLI-17B, by 9.1 points score.

Table 3: Results on Referring Expression Comprehension and Grounded Visual Question Answering.

| Type | Model | RefCOCO | | | RefCOCO+ | | | RefCOCOg | | Visual7W |
|---|---|---|---|---|---|---|---|---|---|---|
| | | val | test-A | test-B | val | test-A | test-B | val | test | test |
| *Generalist* | OFA-L* [Wang et al., 2022b] | 79.96 | 83.67 | 76.39 | 68.29 | 76.00 | 61.75 | 67.57 | 67.58 | - |
| | VisionLLM-H [Wang et al., 2023b] | - | 86.70 | - | - | - | - | - | - | - |
| | Shikra-7B [Chen et al., 2023a] | 87.01 | 90.61 | 80.24 | 81.60 | 87.36 | 72.12 | 82.27 | 82.19 | - |
| | Shikra-13B [Chen et al., 2023a] | 87.83 | 91.11 | 81.81 | 82.89 | 87.79 | 74.41 | 82.64 | 83.16 | 85.33 |
| | Qwen-VL [Bai et al., 2023] | 89.36 | 92.26 | 85.34 | 83.12 | 88.25 | 77.21 | 85.58 | 85.48 | - |
| | Ferret-13B [You et al., 2023] | 89.48 | 92.41 | 84.36 | 82.81 | 88.14 | 75.17 | 85.83 | 86.34 | - |
| | **CogVLM-Grounding** | **92.76** | **94.75** | **88.99** | **88.68** | **92.91** | **83.39** | **89.75** | **90.79** | **91.05** |
| *Specialist* | G-DINO-L [Liu et al., 2023e] | 90.56 | 93.19 | 88.24 | 82.75 | 88.95 | 75.92 | 86.13 | 87.02 | - |
| | UNINEXT-H [Lin et al., 2023a] | 92.64 | 94.33 | 91.46 | 85.24 | 89.63 | 79.79 | 88.73 | 89.37 | - |
| | ONE-PEACE [Wang et al., 2023a] | 92.58 | 94.18 | 89.26 | 88.77 | 92.21 | 83.23 | 89.22 | 89.27 | - |

## 3.2 Visual Question Answering

As illustrated in Table 2, our CogVLM model demonstrates outstanding performance and a significant lead over models of similar parameter scale across a variety of tasks, including daily-life image question-answering dataset VQAv2, text-intensive image question-answering datasets such as TextVQA, and knowledge-demanding datasets like OKVQA and ScienceQA. This success showcases the model's robust generalization capabilities and potential across diverse domains.

## 3.3 LVLM Benchmarks

Our findings, detailed in Table 2, demonstrate that CogVLM achieved state-of-the-art results in all 7 LVLM-benchmarks, markedly surpassing all other models. It also outperformed multimodal models that utilized larger language models, such as LLava1.5 with Vicuna-13B and Emu-2 with LLAMA-33B, leading by 15.7 and 2.6 points on MM-vet, 9.9 and 14.0 points on MMBench, respectively. Compared to IDEFICS-Instruct trained on LLaMA-65B, CogVLM's scores exceeded by 19.3, 23.1, and 20.9 points on Seed-Bench, MMBench, and LLaVA-Bench, respectively. Furthermore, CogVLM achieved a score of 41.1 on the MMMU dataset, and also scored 87.9 on the hallucination assessment dataset POPE, along with 35.2 on the multimodal mathematical reasoning benchmark MathVista. These impressive results not only showcase its robust reasoning abilities and multi-task generalization capabilities but also clearly demonstrate that CogVLM is significantly outpacing other models in these domains. Notably, shallow fusion models such as InstructBLIP and MiniGPT-4 underperformed across most benchmarks, despite InstructBLIP's extensive training on instructional data, underscoring the necessity of deep fusion for enhanced performance.

After using a stronger and larger LLaMA-3 language model as the backbone, our model achieved significant improvements on all benchmarks, fully demonstrating the robustness of our proposed method. The experimental results using other language models as backbones can be found in Appendix C.

Table 4: Ablation studies for various components and training settings. *VE* refers to visual expert.

| Ablated Aspects | Original Setting | Ablated Setting | Trainable params | COCO CIDEr↑ | NoCaps CIDEr↑ | OKVQA top1↑ | TextVQA top1↑ | VQAv2 top1↑ |
|---|---|---|---|---|---|---|---|---|
| Tuned parameters | *VE-full* every layer + MLP Adapter | MLP Adapter | 140M | 131.2 | 111.5 | 55.1 | 40.7 | 73.8 |
| | | LLM+MLP Adapter | 6.9B | 140.3 | 118.5 | 56.8 | 44.7 | 78.9 |
| | | *VE-full* every 4th layer | 1.7B | 138.7 | 117.4 | 58.9 | 44.1 | 77.6 |
| | | *VE-FFN* every layer | 4.4B | 140.0 | 118.7 | 58.2 | 45.1 | 78.6 |
| Init method | From LLM | Random init | 6.6B | 138.0 | 117.9 | 55.9 | 44.0 | 79.1 |
| Visual attention mask | Causal mask | Full mask | 6.6B | 141.0 | 117.2 | 57.4 | 45.1 | 79.6 |
| Image SSL loss | ✗ | ✓(clip feature) | 6.6B | 142.9 | 119.8 | 58.7 | **45.9** | 79.7 |
| Visual encoder | EVA2-E | EVA2-L | 6.6B | 141.4 | **122.5** | 59.2 | 42.8 | 79.0 |
| EMA | ✓ | ✗ | 6.6B | **143.1** | 119.2 | 57.1 | 43.8 | 79.4 |
| *CogVLM (ours)* | — | — | 6.6B | 142.8 | 120.1 | **59.3** | 45.3 | **80.0** |

## 3.4 Visual Grounding

Table 3 shows the result on the standard visual grounding benchmarks. We find that our generalist model achieves state-of-the-art performance across the board, with a significant advantage over the previous or concurrent models. As shown in the bottom part of Table 3, our model even surpasses models that are specifically trained for individual tasks, achieving SOTA performance on 5 of 9 splits. For instance, in the RefCOCO val subset, our model attains a score of 92.76, surpassing UNINEXT-H's 92.64; in the RefCOCO+ test-A subset, it scores 92.91, exceeding ONE-PEACE's 92.21; and in the RefCOCOg test subset, it achieves 90.79, outperforming UNINEXT-H's 89.27. These results suggest a remarkable visual grounding capability of our model incorporating our training paradigm.

## 3.5 Ablation Study

To understand the impact of various components and settings on our model's performance, we conduct an extensive ablation study for 6,000 iterations and a batch size of 8,192. Table 4 summarizes the results about the following aspects:

**Model structure and tuned parameters**. To investigate the effectiveness of CogVLM's model, we conduct ablation studies on several structure variants and tuning strategies, including: 1) tuning only the MLP Adapter layer; 2) tuning all LLM parameters and the Adapter without adding visual expert; 3) only adding visual expert at every 4th LLM layer; and 4) only add visual expert to FFNs at all layers.

From the results, we can see that shallow vision-language alignment, i.e. only tuning the adapter layer (similar to the method used in BLIP-2), results in a significantly inferior performance. Also, the performance of training the visual expert is higher than that of training the LLM, especially on the datasets that require external knowledge, even though the training parameters are roughly the same. We also compare with other variants of adding visual expert, including a. inserting an expert module every 4 layers and b. removing the attention part from the expert. Both of them result in a certain degree of performance decline, but within an acceptable range, which provides some guidance for balancing computational overhead and model performance.

**Initialization Method**. As for visual expert's initialization method, we compare initialization with weights from LLM to random initialization. Our results across various datasets demonstrate that initialization with LLM's weights consistently achieves superior performance. This indicates that the transformer architecture pre-trained on language data possesses a certain capability to process visual tokens. Moreover, it can serve as a more effective starting point for multimodal pre-training initialization.

**Visual Attention Mask**. We empirically find that using a causal mask on visual tokens yields a better result in comparison with a full mask. This is slightly counterintuitive, as using a bidirectional attention mask allows access to more information than a causal mask. We hypothesize the possible explanation for this phenomenon is that the causal mask better fits the inherent structure of LLMs.

**Image SSL Loss**. We also investigated the self-supervised learning loss on image features, where each visual feature predicts the CLIP feature of the next position for visual self-supervision. Align with the observation from PaLI-X [Chen et al., 2023b], we find it brings no improvement on downstream tasks, although we indeed observed improvements in small models in our early experiments.

**Visual Encoder**. we substituted the 300M-parameter EVA2-L model for the 4.4B-parameter EVA2-E to investigate the impact of visual encoder parameters on various tasks. The results indicated that there was only a slight decrease in performance across most benchmarks. However, a notable exception was observed in the text-oriented dataset TextVQA, where we recorded a decline of 2.5.

**EMA**. We utilize EMA (Exponential Moving Average) during pretraining. The ablation results show that EMA often brings improvements across various tasks compared to not using it.

# 4   Conclusion

In this paper, we introduce CogVLM, an open visual language foundation model. CogVLM shifts the paradigm for VLM training from shallow alignment to deep fusion, achieving state-of-the-art performance on 15 classic multi-modal benchmarks.

The VLM training is still in its infancy, and there are many directions to explore, for example, better SFT alignment, RLHF and anti-hallucination. Since the previous famous VLMs are mostly closed-source, we believe CogVLM will be a solid foundation for future multi-modal research.

## 5    Acknowledgments

This work is supported by the Natural Science Foundation of China NSFC 62276148 and 62425601, a research fund from Zhipu, New Cornerstone Science Foundation through the XPLORER PRIZE and Daimler Greater China Ltd. and Tsinghua University Joint Institute for Sustainable Mobility, National Engineering Laboratory for Cyberlearning and Intelligent Technology, and Beijing Key Lab of Networked Multimedia.

## A    Appendix

### A.1    Details of Training Settings

We report the details of parameter settings during pre-training and multitask training in Table 5 and Table 6.

Table 5:  Hyperparameters for pre-training model.

| Hyperparameters | Stage 1 | Stage 2 |
|---|---|---|
| Total steps | $120,000$ | $60,000$ |
| Warmup steps | $12,000$ | $1,200$ |
| Batch size | $8,192$ | $1,024$ |
| Learning rate | $1e^{-4}$ | $1e^{-5}$ |
| Learning rate decay | Cosine | |
| Weight decay | 0.05 | |
| Dropout ratio | 0.1 | |
| Adam $\epsilon$ | $1e^{-8}$ | |
| Adam $\beta$ | (0.9, 0.95) | |
| Textual encoder | Vicuna-1.5-7B | |
| Visual encoder | EVA2-CLIP-E | |
| Patch size | 14 | |
| Input resolution | $224^2$ | $224^2 \rightarrow 490^2$ |

Table 6:  Hyperparameters for multitask finetuning CogVLM.

| Hyperparameters | Multitask |
|---|---|
| Learning rate | $1e^{-5}$ |
| Total steps | 6,000 |
| Batch size | 1,024 |
| AdamW $\epsilon$ | $1e^{-8}$ |
| AdamW $\beta$ | (0.9, 0.95) |
| Weight decay | 0.1 |
| Dropout ratio | 0.1 |
| Input resolution | $490^2$ |

### A.2    Details of Associated Datasets

In this section, we introduce the details of datasets and their use in our evaluation process for all associated benchmarks.

Table 7: Summary of the evaluation benchmarks.

| Task | Dataset | Description | Split | Metrics |
|---|---|---|---|---|
| Image Caption | NoCaps | Captioning of natural images. | val | CIDEr (↑) |
| | Flickr | Captioning of natural images. | karpathy-test | CIDEr (↑) |
| | COCO | Captioning of natural images. | karpathy-test | CIDEr (↑) |
| | TextCaps | Captioning of natural images containing text. | test | CIDEr (↑) |
| General VQA | VQAv2 | VQA on natural images. | test-dev | VQA Score(↑) |
| | OK-VQA | VQA on natural images requiring outside knowledge. | val | VQA Score (↑) |
| | ScienceQA | Multi-choice VQA on a diverse set of science topics | test | Accuracy (↑) |
| | TDIUC | VQA on natural images with detailed question types. | val | VQA Score (↑) |
| LVLM Benchmarks | MM-Vet | Open-ended VQA on a diverse set of topics | test | GPT4 Score(↑) |
| | SEED-Bench | Multi-choice VQA on a diverse set of topics | IMG | Accuracy (↑) |
| | MMBench | Multi-choice VQA on a diverse set of topics | test | Accuracy (↑) |
| | LLaVA-Bench | Open-ended VQA for testing instruction following abilities | In-the-Wild | GPT4 Score(↑) |
| | POPE | Multi-choice VQA for testing hallucinations | overall | Accuracy (↑) |
| | MMMU | VQA on a diverse set of topics | test | Accuracy (↑) |
| | MathVista | VQA for Measuring Mathematical Abilities | test-mini | Accuracy (↑) |
| Grounding | RefCOCO | Refer grounding on natural images. | overall | Accuracy (↑) |
| | RefCOCO+ | Refer grounding on natural images. | overall | Accuracy (↑) |
| | RefCOCOg | Refer grounding on natural images. | overall | Accuracy (↑) |
| | Visual7W | VQA with referential regions selection. | val | Accuracy (↑) |

### A.2.1 Image Captioning

- **COCO** [Lin et al., 2014] The Captions in COCO dataset are collected using Amazon's Mechanical Turk (AMT) workers who are given instructions to control the quality. The dataset contains 330K images, where the train, validation and test sets contain 413,915 captions for 82,783 images, 202,520 captions for 40,504 images, and 379,249 captions for 40,775 images respectively.

- **NoCaps** [Agrawal et al., 2019]. NoCaps is a large-scale benchmark for novel object captioning, containing nearly 400 novel object classes compared to COCO. The validation and test set comprised of 4,500 and 10,600 images, respectively, sourced from the Open Images [Krasin et al., 2017] and annotated with 11 human-generated captions per image, and each set is subdivided into three domains: "in", "near", and "out", with objects in the "out-domain" never appearing in the COCO dataset.

- **Flickr30K** [Plummer et al., 2015]. Flickr30K is a high-quality dataset consists of 31,783 images of everyday life activities, envets and scenes (all harvested from the online website Flickr) and 158,915 captions (obtained via crodsourcing). Each image in this dataset is described independently by five annotators who are not familiar with the specific entities and circumstances depicted in them.

- **TextCaps** [Sidorov et al., 2020] Textcaps is a dataset with 145k captions for 28k images. The design purpose of the TextCaps dataset is to effectively integrate textual information with visual context into captions, requiring the model to have both excellent OCR capabilities and strong captioning abilities.

### A.2.2 General VQA

- **VQAv2** [Antol et al., 2015] VQAv2 encompasses over 200,000 images, paired with more than 1.1 million questions that have collectively garnered over 11 million answers. Questions span various types, including yes/no, counting, and open-ended queries.

- **OKVQA** [Marino et al., 2019] The OK-VQA (Outside Knowledge Visual Question Answering) dataset is specifically designed to probe visual question answering capabilities that necessitate external knowledge or common sense beyond image content. It has 14,055 open-ended questions and 5 ground truth answers per question.

- **ScienceQA** [Lu et al., 2022] The ScienceQA dataset comprises 21,208 multimodal multiple-choice questions spanning three diverse subjects: natural science, language science, and social science. Each question is annotated with explanations linked to relevant lectures.

- **TDIUC** [Shrestha et al., 2019] The TDIUC dataset features 1.6M questions across 170K images from MS COCO and Visual Genome. Categorized into 12 distinct question types, it ranges from basic tasks like identifying objects or colors to more advanced reasoning like counting or positional discernment.

## A.3 LVLM Benchmarks

- **MM-Vet** [Yu et al., 2023] MM-Vet defines six core VL capabilities and examines 16 integrations of interest derived from the combinations of these capabilities. It employs an evaluator based on LLMs for open-ended outputs, capable of assessing across different question types and answer styles, thus deriving a unified scoring metric.

- **SEED-Bench** [Li et al., 2023a] SEED-Bench is a dataset comprising 19K multiple-choice questions with precise human annotations, covering 12 evaluation dimensions, including understanding of image and video modalities. It obtains accurate answer options through manual annotations, enabling objective and efficient assessment of model performance.

- **MMBench** [Liu et al., 2023g] MMBench comprises approximately 3000 multiple-choice questions, covering 20 different capability dimensions, aimed at evaluating various abilities of visual-language models. MMBench adopts a hierarchical capability dimension structure, including two high-level capability dimensions: perception and reasoning, as well as fine-grained capability dimensions such as object localization and attribute inference.

- **LLaVA-Bench** [Liu et al., 2023c] LLaVA-Bench (In-the-Wild) is a benchmark dataset comprising 60 questions, designed to evaluate the multimodal instruction following capabilities of LMMs. It includes indoor and outdoor scenes, memes, paintings, sketches, etc., and is equipped with highly detailed, manually curated descriptions and appropriate question selections.

- **POPE** [Li et al., 2023c] The POPE dataset is a binary classification query dataset specifically designed to evaluate object hallucination issues in LMMs. The random, popular, and adversarial subsets within the POPE dataset are constructed through different sampling strategies, totaling 8,910 entries.

- **MMMU** [Yue et al., 2023] The MMMU dataset is a large-scale, multidisciplinary multimodal understanding and reasoning benchmark set, containing 11.5K questions. It covers 6 major disciplines, 30 topics, and 183 subfields, with question types including multiple-choice and open-ended questions. The dataset includes 30 types of images, such as charts, tables, chemical structures, photographs, paintings, musical scores, etc., testing the multimodal perception capabilities of models and their performance in expert-level tasks.

- **MathVista** [Lu et al., 2023] MathVista is a new benchmark dataset that combines mathematical and visual understanding, comprising 31 existing multimodal datasets and 3 newly created datasets, totaling 6141 examples. These datasets encompass a diverse range of mathematical reasoning abilities, including seven types: algebra, arithmetic, geometry, logic, numerical common sense, science, and statistics. The goal is to comprehensively evaluate the capabilities of existing foundational models in mathematical reasoning and visual understanding.

### A.3.1 Grounding

- **RefCOCO/RefCOCO+** [Liu et al., 2017] RefCOCO and RefCOCO+ evolved from the ReferItGame. Both subsets focus on images with two or more similar objects. RefCOCO, with 142,209 expressions across 19,994 images, places no linguistic constraints. Conversely, RefCOCO+ emphasizes appearance-centric descriptions, omitting locational terms, and comprises 141,564 expressions over 19,992 images.

- **RefCOCOg** [Mao et al., 2016] The RefCOCOg subset was amassed through Amazon Mechanical Turk, where workers penned natural referring expressions for objects in MSCOCO images; it boasts 85,474 referring expressions spanning 26,711 images, each containing 2 to 4 objects of the same category.

- **Visual7W** [Zhu et al., 2016]. The Visual7W dataset is predominantly designed for VQA tasks, with a dedicated subset crafted for grounded VQA. In this subset, models are presented with an image accompanied by a "which"-type question, such as "Which is the small computer in the corner?". Participants are then given four bounding boxes within the image, from which they must select the correct one as the answer. The grounded Visual7W part consists of 25,733 images and 188,068 questions.

- **Flickr30K-Entities** [Plummer et al., 2015]. The Flickr30K Entities dataset, a precursor in the realm of grounded captioning, encompasses a collection of 31,783 images accompanied

by 158k captioning annotations. Every caption in this dataset has been meticulously annotated such that each noun phrase is linked with a manually delineated referential bounding box. In total, there are 276k such annotated bounding boxes provided within this dataset.

- **VisualGenome [Krishna et al., 2017].** The VisualGenome dataset stands as a cornerstone in understanding the multifaceted relationships present within images. With a collection of over 100k images, each image is annotated in detail, capturing an average of 21 objects, 18 attributes, and 18 inter-object relationships. A unique aspect of this dataset is the alignment of objects, attributes, relationships, and region descriptions with standardized terminologies from WordNet. Specifically tailored for the REG and REC tasks, each annotated region in an image comes with a corresponding descriptive text, making it a rich resource for image understanding and semantic modeling. We use the subset with around 86k images and 3.6 million region-caption pairs for visual grounding.

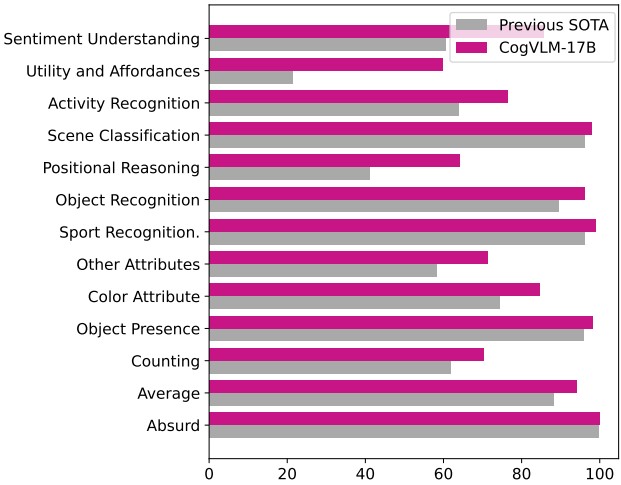

Figure 4: Performance on TDIUC benchmark with fine-grained questions classes.

## B    Additional Fine-grained Experiments

To comprehensively investigate the proposed model on specific topics and question types, we further conduct extensive experiments on a representative benchmark, TDIUC [Kafle and Kanan, 2017]. We use the publicly available split of val set as evaluation data, and the VQA accuracy calculated from their official scripts as the evaluation metric.

The experimental results on TDIUC compare our model against the specialist SOTA method MUREL [Cadene et al., 2019] are shown in Figure 4. From the experimental result, we can see that our model consistently outperforms the previous model on 12 specific question types, resulting in a 94.0 accuracy score compared to the previous SOTA of 88.2 on the overall dataset. These results demonstrate that our model exhibits comprehensive problem-solving skills on general VQA tasks.

## C    Alternative Language Models Results

Table 8: Comparison of different language models as backbones.

| LLM | MM-Vet | OKVQA | MathVista | MMBench |
|---|---|---|---|---|
| Vicuna-7B-1.5 | 52.0 | 64.8 | 34.5 | 77.6 |
| Vicuna-13B-1.5 | 56.8 | 66.7 | 37.2 | 78.1 |
| LLaMA3-8B | 60.4 | 64.1 | 38.1 | 80.5 |
| GLM3-32B | 64.5 | 68.2 | 45.1 | 82.3 |

As shown in the Table 8, our visual expert module is integrated into the LLM, and it can significantly benefit from the scaling of the LLM. These results demonstrate that our approach can effectively leverage the benefits of LLM scaling to improve performance on multimodal tasks.

## D  Computational Efficiency

In this section, we compare the computational efficiency of our model with other state-of-the-art models, considering both pretraining and finetuning data from datasets such as VQAv2 and TextVQA. Owing to an optimized architecture and the utilization of high-quality pretraining data, our model demonstrates a marked reduction in resource consumption during training relative to models with comparable parameter magnitudes.

Table 9: Comparison of different models based on their computational efficiency. We use PFLOPS*days as metrics.

| Model | Pretraining Data | Pretraining compute | VQAv2 finetuning | TextVQA finetuning |
|---|---|---|---|---|
| PaLI-3B | 1.6B | 56 | 1.1 | 0.2 |
| PaLI-17B | 1.6B | 453 | 4.5 | 0.9 |
| Flamingo-80B | 2.3B | 1381* | N/A | N/A |
| GIT2-5.1B | 12.9B | 5513* | N/A | N/A |
| CogVLM | 1.5B | 230.1 | 1.2 | 0.13 |

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
