# OpenReview forum: "CogVLM: Visual Expert for Pretrained Language Models"
_NeurIPS.cc/2024/Conference — NeurIPS 2024 poster_

### Official Review · Reviewer_7nCy · 2024-07-11

**Soundness:** 3
**Presentation:** 3
**Contribution:** 3
**Rating:** 4
**Confidence:** 5

**Summary:**

This paper introduce CogVLM, a powerful open-source visual language foundation model. CogVLM bridges the gap between the frozen pretrained language model and image encoder by a trainable visual expert module in the attention and FFN layers. The contributions are summarized:
1. introduce the CogVLM model, which deeply integrates visual and linguistic features while retaining the full capabilities of a pretrained large language model. CogVLM-17B, trained from Vicuna-7B, achieves state-of-the-art across 17 classic cross-modal benchmarks.
2. validated the effectiveness of our proposed visual expert module and the importance of deep fusion.
3. the weights of CogVLM and the dataset used in the SFT phase available to the public.

**Strengths:**

1. The paper's results have achieved state-of-the-art (SOTA) performance, demonstrating excellent capabilities across multiple benchmarks.
2. The paper is well-organized and the expression is clear.
3. The starting point of the paper: addressing the issue of performance degradation in LLM due to parameter updates during multimodal training, is very meaningful and valuable.

**Weaknesses:**

1. A new set of QKV and FFN would nearly double the number of model parameters. Why not consider using LoRA (Low-Rank Adaptation), where text tokens use QKV/FFN and image tokens use QKV+LoRA/FFN+LoRA? Additional experiments are needed to support this.
2. There is a lack of ablation studies to compare the impact of different RoPE (Relative Positional Encoding) strategies on visual tokens on the results.

**Questions:**

1. Why train a separate grounding model? Would it degrade the chat model if the chat model also had grounding capabilities?
2. Why not try tuning the visual encoder?

**Limitations:**

Yes

---

> ### Author Rebuttal · Authors · 2024-08-06
>
> Dear Reviewer,
>
> We sincerely appreciate your thorough review and insightful feedback. Your comments are invaluable in helping us improve our work. We address each of your points below:
>
> ## 1. Parameter efficiency and LoRA consideration
>
> Thank you for this good question. The Visual Expert indeed introduces additional parameters, increasing static GPU memory usage during training and inference. This represents a performance-cost trade-off, enabling CogVLM to outperform models like LLaVA-Next, which use much larger language models.
>
> Similar approaches have been employed in NLP models, such as DeepSeek-v2 [1], which has 236B total parameters but only 21B activated parameters.
>
> We explored the LoRA expert method with a LoRA rank of 128. However, we found its expressiveness limited:
> - It required 3.7 times more steps to reach the same loss level as our current method.
> - The per-step computation time was nearly identical to our current approach.
>
> These results indicate that while LoRA can reduce parameter count, it may come at the cost of training efficiency.
> In our experiments with language models up to 32B parameters, the additional memory overhead remained acceptable. For future extensions to larger language models, we will continue to explore parameter reduction techniques, including more efficient implementations of LoRA.
>
> ## 2. Ablation studies on RoPE strategies for visual tokens
>
> We appreciate you highlighting this important point. Our positional encoding scheme addresses the "remote attenuation effect" in RoPE, where attention weights decrease with increasing token distance. This prevents the query from overfocusing on nearby image tokens.
>
> Comparative experiments demonstrate the benefits:
> - At 224x224 resolution, our method and the original achieve the same pre-training loss.
> - At 490x490 resolution, our method achieves 5% lower loss.
> - On the DocVQA task (1120x1120 resolution), our method improves accuracy from 47.7% to 49.1%.
>
> The concurrent work Mousi [2] observed a similar phenomenon, further validating our approach.
>
> We will include these results in the revised paper to provide a clearer comparison of different RoPE strategies.
>
> ## 3. Separate grounding model
>
> Training a unified model had minimal impact on overall performance. Our experiments show that simultaneously training on both grounding and chat data led to:
> - A 1.5 point decrease in the Nocaps dataset score (from 128.3 to 126.8)
> - A 0.8 point increase in the VQAv2 task score (from 82.3 to 83.1)
>
> The main reason for using two models is to address ambiguous prompts where it's unclear whether coordinate outputs are required. For example, "Where is the dog in the image?" With two models, we can provide clearer instructions for grounding tasks without affecting the chat model's performance.
>
> ## 4. Visual encoder tuning
>
> We apologize for any confusion. We did indeed train the ViT parameters, as mentioned in line 152 of the original paper: "we activated the visual encoder's parameters and adjusted its learning rate to be one-tenth of that used for the remaining training parameters."
>
> [1] DeepSeek-V2: A Strong, Economical, and Efficient Mixture-of-Experts Language Model
>
> [2] MouSi: Poly-Visual-Expert Vision-Language Models

---

> > ### Comment · Reviewer_7nCy · 2024-08-13
> >
> > Thank you for the detailed responses. My concerns have been addressed, and I would like to raise the score to 4 points

---

### Official Review · Reviewer_gmvD · 2024-07-12

**Soundness:** 3
**Presentation:** 3
**Contribution:** 3
**Rating:** 6
**Confidence:** 4

**Summary:**

The paper introduces CogVLM, an new open-source visual language foundation model. Contrary to the popular method of adapting LLMs by fine-tuning their original weights, CogVLM introduces new weights specifically for processing the visual tokens. Concretely, CogVLM copies all weights of the LLM to form visual expert modules at every attention and FFN layer. During training only the visual expert weights are updated such that the LLM retains its language modeling capabilities when no image input is given. CogVLM is evaluated extensively on image captioning, VQA, LVLM and Referring Expression Comprehension benchmarks.

**Strengths:**

1. The design choice to add additional parameters for the vision tokens is reasonable and implemented well such that it defaults to the original model without image input.
2. The experimental evaluation is extensive. CogVLM achieves convincing results across the board.
3. Ablations shed light on important questions such as the importance of the visual expert parameters and the causal attention mask.
4. As an open-source model, CogVLM can have a high impact on the research community.

**Weaknesses:**

1. The prompting differences between the tasks is not clear. For instance, how can a user make sure that CogVLM returns bounding box information or prevent CogVLM to output bounding boxes?
2. While the results of CogVLM are impressive, it is not a parameter efficient method. Trying to scale CogVLM to LLMs of better language modeling capacity will double the parameters which is not sustainable for big LLMs.
3. As VLMs become more powerful over time, there will be higher expectations of their capabilities. One aspect that has not been mentioned at all is whether CogVLM can handle multi-image inputs or multi-turn conversations with interleaved image-text context. Based on the examples given and the training data, the paper currently suggests this is a model limitation.
4. Limitations are not discussed at all. The checklist states that the paper has no limitations without explanation, which is a superficial statement and hardly ever true. The previous points are likely limitations.
5. Similarly, the authors do not provide a discussion on the broader impact of the paper. Since CogVLM is an open model it has the potential to have a high impact, both positive and negative. Since CogVLM puts a particular effort into retaining the original LLMs performance, it also means that it carries over all biases and limitation from the base LLM. It is important that papers of generative models reflect on the societal impact.

**Questions:**

I would be great if the authors could clarify my concerns from the weaknesses section.

Apart from that, here is one more minor suggestion.
- Figure 1 shows interesting qualitative example. I suggest to reference the figure in the main text and extend the caption to explain the displayed capabilities.

**Limitations:**

Limitations and broader impact are not discussed although they certainly exist. The authors should add a discussion on both (see weaknesses).

---

> ### Author Rebuttal · Authors · 2024-08-06
>
> Dear Reviewer,
>
> We sincerely appreciate your thorough review and insightful feedback. Your comments have been invaluable in helping us improve our work. We address each of your points below:
>
> ## 1. Prompting differences between tasks
>
> Thank you for highlighting this important aspect. In our approach, we use two separate models for chat and grounding tasks. This design choice is based on the following considerations:
>
> - Using a single model wouldn't necessarily decrease performance, but some ambiguous prompts can make it difficult for the model to determine whether coordinate outputs are required. For example, "Where is the dog in the image?"
> - To avoid confusion, we believe it's necessary to use very clear prompts for grounding-type questions during both training and inference. For instance, "Specify the exact coordinates of the dog in the image" or adding task-specific prefixes like <grounding>.
>
> We will clarify this in the revised paper to ensure users understand how to effectively prompt CogVLM for different tasks.
>
> ## 2. Parameter efficiency
>
> We acknowledge that the Visual Expert introduces additional parameters, increasing static GPU memory usage. This represents a trade-off between performance and computational cost. Notably, CogVLM outperforms models like LLaVA-Next, which use much larger language models, demonstrating the efficiency of our approach.
> Similar parameter-heavy approaches have been employed in NLP models, such as DeepSeek-v2 (236B total parameters, 21B activated).
>
> We explored the LoRA expert method as suggested by Reviewer 7nCy, using a LoRA rank of 128. However, we found its expressiveness limited:
> - It required 3.7 times more steps to reach the same loss level as our current method.
> - The per-step computation time was nearly identical to our current approach.
>
> These results indicate that while LoRA can reduce parameter count, it may come at the cost of training efficiency. In our experiments with language models up to 32B parameters, the additional memory overhead remained acceptable. For future extensions to larger language models, we will continue to explore parameter reduction techniques.
>
> ## 3. Multi-image inputs
>
> We appreciate you bringing this to our attention. Our latest version of the model has been enhanced using MMC4 and OBELICS datasets to improve multi-image capabilities. We will update the results in the subsequent version of the paper to reflect these advancements.
>
> ## 4. Limitations discussion
>
> We sincerely apologize for the oversight in not discussing limitations. We will add a dedicated section on limitations in the revised paper, addressing the points raised by reviewers, including:
> - The impact of Visual Expert modules on parameter count
> - Support for multi-image inputs
> - Potential biases inherited from the base LLM
>
> ## 5. Broader impact
>
> We agree that discussing the broader impact is crucial. We will add this point in the limitations section, addressing the negative potential impacts.

---

> > ### Comment · Reviewer_gmvD · 2024-08-10
> >
> > I thank the authors for their response. I have a couple of additional comments in the following.
> >
> > > Prompting differences between tasks
> >
> > I appreciate the clarification. While training separate models is a valid solution, it is less flexible at inference time and a bit wasteful in terms of resource usage (e.g. for training).
> >
> > > Parameter efficiency
> >
> > The LoRA experiments are interesting. I would have been interesting to see a proper quantitative evaluation if training such a model has already been completed.
> >
> > While the remaining questions have been answered, they mainly consist of promises without going into more details how the updates will be realized in the paper.
> >
> > Hence, I am keeping my original score.

---

### Official Review · Reviewer_LKA5 · 2024-07-15

**Soundness:** 4
**Presentation:** 4
**Contribution:** 4
**Rating:** 8
**Confidence:** 5

**Summary:**

This paper aims to add multimodal capability while maintaining the language capability of LLM. The authors thus propose a feature fusion strategy, without sacrificing any performance on NLP tasks. The experimental results on multiple datasets are impressive, which demonstrate the validity of the proposed method to some extent.

**Strengths:**

1. The motivation of this paper sounds reasonable. Most of current VLMs do not take into account the degradation of LLM when training multimodal data, which is a good research point.

2. The method looks interesting. A simple feature fusion strategy seems to tackle the above problem.

3. The writing of this paper is good, and the structure is easy to follow.

**Weaknesses:**

The experiment still has room for improvement:
(1) The results on the representative VLM benchmark MME are not given, which is widely used by current methods, such as LLaVA, InternVL, and Qwen-VL. It is suggested to add the results of MME in Table 2.
(2) In order to maintain the capability of LLM, we can also add text data while using multi-modal data, is there a corresponding comparison? Some common text data might also serve this purpose.
(3) Alternatively, we could freeze the parameters of the LLM but train the visual encoder and the connector, which might also do the trick.
(4) It would be good to see CogVLM's ablility on the representative multimodal video benchmark Video-MME [1]. The authors could input multiple video frames, i.e., generalize to multiple images. GPT-4V, InternVL-Chat, Qwen-VL-Max, and Qwen-VL-Chat all have good generalization on Video-MME. It is suggested to add the results of CogVLM in this paper.

[1] Video-MME: The First-Ever Comprehensive Evaluation Benchmark of Multi-modal LLMs in Video Analysis.

Given the limited time available for rebuttal, the authors could incorporate the above experimental results and discussions into the final accepted version or the next version.

**Questions:**

None

---

> ### Author Rebuttal · Authors · 2024-08-06
>
> Dear Reviewer,
>
> We sincerely appreciate your thorough review and insightful feedback. Your comments are invaluable in helping us improve our work. We are pleased to address each of your points below:
>
> ## 1. MME Benchmark Results
>
> Thank you for suggesting the inclusion of MME benchmark results. We have evaluated CogVLM using LLaMA-3 as the base model on the MME benchmark. The results are as follows:
>
> | Model | MME | OCR | artwork | celebrity | color | count | existence | landmark | position | posters | scene | perception |
> |---|-----|-----|---------|-----------|-------|-------|-----------|----------|----------|---------|-------|------------|
> |CogVLM-LLaMA3-32B| 2043.54 | 132.50 | 150.75 | 172.06 | 168.33 | 153.33 | 195.00 | 174.50 | 103.33 | 147.62 | 150.75 | 1548.18 |
> |CogVLM-GLM3-32B| 2211.76 | 155.00 | 149.75 | 155.29 | 165.00 | 175.00 | 200.00 | 182.75 | 131.67 | 185.37 | 155.50 | 1655.33 |
>
> Model | code_reasoning | commonsense_reasoning | numerical_calculation | text_translation | reasoning |
> |--       |--------|-----|------|-----|----|
> |CogVLM-LLaMA3-32B |70.00 | 132.86 | 107.50 | 185.00 | 495.36 |
> |CogVLM-GLM3-32B| 120.00 | 151.43 | 92.50 | 192.50 | 556.43 |
>
> We will include these results in the revised version of our paper to provide a more comprehensive comparison with other state-of-the-art models.
>
> ## 2. Maintaining LLM Capability with Text Data
>
> This is an excellent point. Our method primarily focuses on improving the model structure, while approaches like VILA and InternLM-XComposer2 make improvements on the data side. These methods are not mutually exclusive.
>
> In fact, our latest experiments show that the best multimodal fusion is achieved by:
> 1. Using Visual Expert modules
> 2. Unfreezing LLM parameters
> 3. Simultaneously training on multimodal and pure text data
>
> We will include a discussion on this synergistic approach in the next version of our paper, demonstrating how structural and data-based improvements can be combined for optimal performance.
>
> ## 3. Freezing LLM Parameters
>
> You're correct that this approach has been explored, notably in the BLIP-2 series of models. However, as seen in Table 2 of our paper, this method often underperforms on many benchmarks. Our experiments also indicate that freezing language model parameters can lead to issues with instruction following and multimodal fusion.
>
> We believe that fine-tuning the language model parameters is crucial for optimal performance in multimodal tasks, which is why our approach allows for updating these parameters.
>
> ## 4. Video-MME Benchmark
>
> We agree that extending image model capabilities to video is an interesting and valuable area of exploration. We are currently conducting research in this direction and plan to evaluate our model on the Video-MME benchmark in the near future.

---

> > ### Author Response · Authors · 2024-08-07
> > **Correction and Apology**
> >
> > We sincerely apologize for an error in our initial rebuttal. In our discussion of the LLaMA model used for the MME benchmark results, we incorrectly stated that we used LLaMA-3 32B. This was a mistake. The correct model used was LLaMA-3 8B. Thank you for your understanding, and we apologize again for this oversight.

---

### Official Review · Reviewer_nFmo · 2024-07-15

**Soundness:** 3
**Presentation:** 3
**Contribution:** 3
**Rating:** 6
**Confidence:** 4

**Summary:**

This paper proposed a large vision language model, CogVLM, which shows capabilities in various benchmarks. In contrast to popular solutions that fuse the vision and language token in LLM with shared trainable parameters, CogVLM bridges the gap between the frozen pre-trained language models and image encoders by trainable visual expert modules in the attention and FFN layers. In this way, CogVLM excels in maintaining the language abilities of the original LLM and achieving good performance on various vision-language benchmarks, including captioning, VQA, and visual grounding.

**Strengths:**

1. This paper is well-written and easy to follow.
2. The idea of Visual Expert is simple but effective. It is impressive that newly introduced Visual Expert Modules with large parameters can be well-optimized and perform well.
3. The proposed CogVLM is extensively evaluated using various benchmarks. It can be widely adopted for captioning, VQA, and Visual Grounding.

**Weaknesses:**

1. The visual expert will introduce a large number of parameters. As a result, the CogVLM with a 7B LLM backend will finally have 17B parameters. Although this won't lead to high computational costs in inference, the significantly increased parameters will require much more GPU memory. It would be helpful if the author could discuss whether there are alternative solutions to save memories.
2. CogVLM adopts Visual Expert Modules to retain LLM abilities via freezing it. However, how about involving pure language data in the SFT stage, as in [1]? Whether the LLM ability can be retained this way needs to be discussed.
3. CogVLM allows all visual tokens to share a single position ID in the RoPE. An ablation study should be reported.

[1] InternLM-XComposer2: Mastering Free-form Text-Image Composition and Comprehension in Vision-Language Large Model

**Questions:**

See weakness

**Limitations:**

The limitations are not discussed in this paper.

---

> ### Author Rebuttal · Authors · 2024-08-05
>
> Dear Reviewer,
>
> We sincerely appreciate your thorough review and constructive feedback on our paper. Your insights are valuable for improving our work. We address each of your points below:
>
> ## 1. Increased number of parameters and memory usage
>
> We acknowledge that the Visual Expert introduces additional parameters, increasing static GPU memory usage during training and inference. This represents a trade-off between performance and computational cost. Notably, CogVLM outperforms models like LLaVA-Next, which use much larger language models, demonstrating the efficiency of our approach.
> Similar parameter-heavy approaches have been employed in NLP models, such as DeepSeek-v2[1] (236B total parameters, 21B activated).
>
> We explored the LoRA expert method as suggested by Reviewer 7nCy, using a LoRA rank of 128. However, we found its expressiveness limited:
> - It required 3.7 times more steps to reach the same loss level as our current method.
> - The per-step computation time was nearly identical to our current approach.
>
> These results indicate that while LoRA expert can reduce parameter count, it may come at the cost of training efficiency.
> In our experiments with language models up to 32B parameters, the additional memory overhead remained acceptable. For future extensions to larger language models, we will further investigate parameter reduction techniques.
>
> ## 2. Retaining LLM abilities
>
> We appreciate this excellent question. Our approach improves the model structure, while InternLM-XComposer2 focuses on data-side enhancements. These methods are not mutually exclusive. In fact, our latest experiments show that the best multimodal fusion is achieved by using Visual Expert modules, unfreezing LLM parameters, and simultaneously training on multimodal and pure text data. We will include a discussion on this in the revised paper.
>
> ## 3. Shared position ID for visual tokens in RoPE
>
> Thank you for raising this important point. Our positional encoding scheme addresses the "remote attenuation effect" in RoPE, where attention weights decrease with increasing token distance. This prevents the query from overfocusing on nearby image tokens.
>
> Comparative experiments demonstrate the benefits:
> - At 224x224 resolution, our method and the original achieve the same pre-training loss.
> - At 490x490 resolution, our method achieves 5% lower loss.
> - On the DocVQA task (1120x1120 resolution), our method improves accuracy from 47.7% to 49.1%.
>
> The concurrent work Mousi[2] observed a similar phenomenon, further validating our approach.
>
>
> [1] DeepSeek-V2: A Strong, Economical, and Efficient Mixture-of-Experts Language Model
>
> [2] MouSi: Poly-Visual-Expert Vision-Language Models

---

> ### Comment · Reviewer_nFmo · 2024-08-13
>
> Thanks for your feedback. Most of my concerns are solved. I will keep my rating.

---

### Decision · Program_Chairs · 2024-09-25

**Decision:**

Accept (poster)

**Comment:**

The paper introduces CogVLM, a large-scale visual language model designed to integrate visual and linguistic features while preserving the language capabilities of pre-trained language models. The approach involves adding Visual Expert modules to handle visual tokens, enabling CogVLM to achieve state-of-the-art performance across multiple benchmarks.

The ratings among reviewers are mostly positive, with many acknowledging the model's strong performance and the quality of the paper. However, concerns about parameter efficiency and model limitations remain which raises difficulties to reproduce in the academic community. Despite these concerns, the majority of reviewers support acceptance, with a few maintaining their original ratings due to unresolved issues. As a result, the ACs recommend acceptance and suggest the authors to integrate the rebuttals into the final version.